# Optimized High Resolution 3D Dense-U-Net Network for Brain and Spine Segmentation [†]

**Martin Kolařík [1,*], Radim Burget [1], Václav Uher [1], Kamil Říha [1] and Malay Kishore Dutta [2]**

[1] Department of Telecommunications, Brno University of Technology, Brno 616 00, Czech Republic; burgetrm@feec.vutbr.cz (R.B.); xuherv00@stud.feec.vutbr.cz (V.U.); rihak@feec.vutbr.cz (K.Ř.)

[2] Centre for Advanced Studies, Dr. A.P.J. Abdul Kalam Technical University, Lucknow 226031, India; malaykishoredutta@gmail.com

[*] Correspondence: martin.kolarik@vutbr.cz or xkolar54@stud.feec.vutbr.cz

[†] This paper is an extended version of our paper published in 3D Dense-U-Net for MRI Brain Tissue Segmentation Published in 2018 41st International Conference on Telecommunications and Signal Processing (TSP).

**Abstract:** The 3D image segmentation is the process of partitioning a digital 3D volumes into multiple segments. This paper presents a fully automatic method for high resolution 3D volumetric segmentation of medical image data using modern supervised deep learning approach. We introduce 3D Dense-U-Net neural network architecture implementing densely connected layers. It has been optimized for graphic process unit accelerated high resolution image processing on currently available hardware (Nvidia GTX 1080ti). The method has been evaluated on MRI brain 3D volumetric dataset and CT thoracic scan dataset for spine segmentation. In contrast with many previous methods, our approach is capable of precise segmentation of the input image data in the original resolution, without any pre-processing of the input image. It can process image data in 3D and has achieved accuracy of 99.72% on MRI brain dataset, which outperformed results achieved by human expert. On lumbar and thoracic vertebrae CT dataset it has achieved the accuracy of 99.80%. The architecture proposed in this paper can also be easily applied to any task already using U-Net network as a segmentation algorithm to enhance its results. Complete source code was released online under open-source license.

**Keywords:** 3D segmentation; brain; deep learning; neural network; open-source; semantic segmentation; spine; u-net

## 1. Introduction

Image segmentation is an important process of automated image processing based on the principle of partitioning the input image into areas sharing common features and therefore extract the information that the input image contains. The segmentation itself can be described as a method for labelling each pixel, or in the case of 3D data each voxel, with a corresponding class and is used nowadays as one of the basic image processing method for understanding the content of the input image in many areas of the computer vision.

Together with a rising availability of modern medical image scanning systems such as the magnetic resonance imaging (MRI) or computed tomography (CT) comes the need for automated processing of the scanned data. Evaluation of the results gathered by these scanning methods is usually done by hand by doctors and can be a repetitive and time–consuming task even for an experienced radiologist. The automation of this process is therefore very valuable and can help doctors to determine the correct diagnose faster when they are presented with precisely segmented scanned data within few seconds.

Problem with automatic segmentation of any tissue in medicine is that the method must be reliable and at least as precise as when doctor would have done the same task by hand.

The majority of work in medical image data segmentation focuses on segmenting abnormal tissue regions to determine correct diagnose and the progress of cancer tumours. The problem we are addressing with this paper is the semantic segmentation and subsequent modelling of 3D volumetric segmentations of brain and spine of the patient from MRI and CT scans. Example of MRI and CT scan slices can be seen in Figure 1. Current technology in additive manufacturing and virtual reality brings the doctors new possibilities in examining the patient before operation in high level of detail. To help automate the process of creating accurate 3D models of different parts of human body we propose an optimized neural network architecture evaluated on both MRI and CT images of soft and bone tissue capable of processing data in its original resolution and accelerated on graphical process unit (GPU) for faster parallel computation.

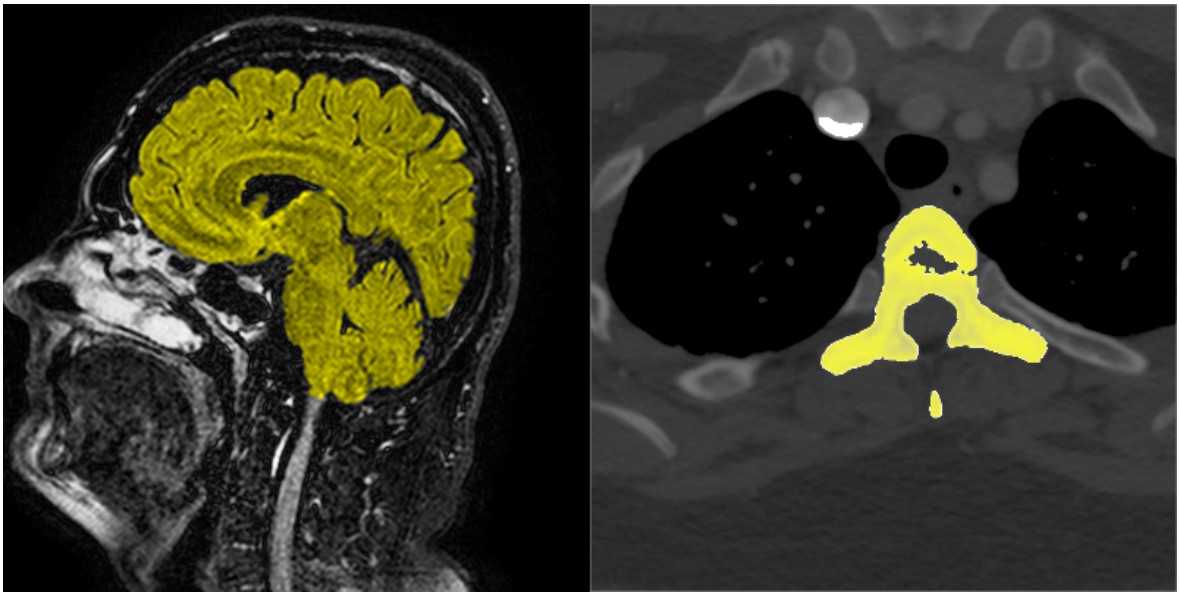

**Figure 1.** Example of MRI sagittal brain scan slice (**left**) and CT transversal thoracic scan slice (**right**)—tissue segmented with our system is highlighted in yellow.

The Paper is structured into four main sections—Introduction, Materials and Methods, Results and Conclusions. We also added Appendix A. in which we describe how to implement our published source code to segment image data other than the datasets used in this paper. Section 1 covers Introduction and Related works in the field of segmentation in general and also in focus with segmenting medical image data. We also describe specifics of segmenting the brain and spine image data. In the Section 2—Materials and Methods, we describe used datasets, neural network architectures and training process. This section provides all the information needed in order to replicate our experiment. In the Section 3, Results, we publish the results of our experiment measured in five different metrics and also provide visualisation of segmented 3D models of brain and spine created by our algorithm. In the Section 4, Conclusions, we summarize the results of this paper and provide future ways where we want to aim our work in medical image segmentation using deep densely conected neural networks.

*1.1. Background and Related Work*

1.1.1. Deep Learning Segmentation Methods

Current state of the art methods for segmenting image data are deep learning based. A comparison of most currently used network architectures is described in [1]. Shelhamer et al. [2] proposed the first

fully convolutional architecture for segmentation. This work laid foundations for most currently used segmentation neural network architectures, because using only convolutional layers made a significant rise in accuracy. However, this architecture needs a large dataset to be fully trained and therefore it is not usually suitable for medical image segmentation. Using only convolutional layers combined with skip connections became one of the most popular architectures the autoencoder type of network called U-Net [3]. It consists of downsampling, bridge and upsampling blocks and is widely used for segmenting medical data thanks to its ability to be properly trained using little data comparing to its competitors such as [2]. Development in the area of classification networks using data interconnections from coarse to fine layers of the network as in residual networks [4] and followed by densely connected networks [5] led to application of this principle also in segmentation networks. Popular segmentation architecture using these principles is Tiramisu network [6]. All cited architectures are designed for 2D data. Reason for 2D data processing is that these networks were primarily published to be used on general image data and also because the training is implemented for GPU. GPU is used for parallel data processing and can achieve 20 or more times faster computation, but with limited memory use. With the latest advancements in GPU technology we are able to train larger architectures due to increased GPU RAM memory and utilize information in the third dimension for higher accuracy, because most medical data that are used in clinical practice consist of 3D image volumes.

Implementation of U-Net network for 3D data processing was done in [7] and achieved higher accuracy on testing data than original U-Net. Another design of 3D segmentation U-Net type network was done in [8]. Combining 2D and 3D data processing in a hybrid densely connected neural network architecture was done in [9] These methods however are designed to process medical images in lower resolution and therefore are not suitable for processing high resolution thoracic and other upper body medical images. The implementation and evaluation of residual and dense interconnections in the 3D U-Net segmentation model is the main proposal of this paper. We tested both Residual-U-Net and Dense-U-Net architectures and optimized them for medical image processing in 3D.

In contrast with previously mentioned architectures, our goal was to design U-Net architecture with densely connected layers for 3D data processing optimized for processing data in high resolution and compare its accuracy with the results of the original U-Net implementation and U-Net with added residual connections.

### 1.1.2. Brain Segmentation

The problem of automatic brain tissue segmentation has been very well explored before deep learning algorithms became a standard for semantic segmentation. Despotović et al. [10] covers an overview of older methods such as thresholding, clustering or some form of simpler machine learning algorithms. These methods also rely heavily on image preprocessing as in [11] therefore they are not automatic. One of quite frequently used preprocessing method is some form of skull-stripping algorithm, which removes the bone tissue of skull from the input picture before processing as in [12]. One of the most notable works in the brain segmentation field is the FMRIB's Automated Segmentation Tool (FSL) [13]. This method is based on a hidden Markov random field model and an associated Expectation-Maximization algorithm. We used this tool to evaluate our brain segmentation results to a method that does not use deep neural networks. Since deep learning methods achieve higher accuracy even without any preprocessing of input image, most of the ongoing research including this paper is now using some form of deep neural network for brain segmentation. An overview of the current state-of-the-art in the field of brain segmentation using deep learning can be found in [14]. This is an extended version of our paper 3D Dense-U-Net for MRI Brain Tissue Segmentation which was focused only on brain segmentation [15].

### 1.1.3. Spine Segmentation

Segmentation and volumetric 3D modelling of individual vertebrae or the complete human spine is an important task for surgeon pre-operation preparation. Combined with 3D printing or

virtual reality systems for spine model examination, the surgeon gets much deeper understanding of the patient's problem. As the input images are usually thoracic scans, simple methods like thresholding are unable to distinguish between all bones in the thoracic region and spine segmentation is a difficult task. Spine segmentation is a challenging problem also because the input images contain less distinctive features compared to brain segmentation. Also when scanning younger children with not fully ossified bones, the contrast is very low and the spine tissue is not easily distinguishable from surrounding tissues.

Recent work using deep learning U-Net architecture [16] achieved highly precise results and by combining more neural network architectures in a chain even a higher precision can be achieved [17].

## 2. Data and Segmentation Methods

This section covers overview and implementation details of used segmentation architectures. Subsection dataset contains all the information about data we used for this paper and their preparation process.

### *2.1. Dataset*

Modern convolutional deep learning architectures require large and properly annotated datasets divided into three parts—training, validation and testing data. Training is the part of the dataset that is used for learning the feature representation of the input data. Validation data are used during training process to control the progress of training accuracy. Testing data are used for validating accuracy of the algorithm after the training process. They are not used during the training process and should represent the production data which the algorithm will be used for.

### 2.1.1. Medical Image Data Formats

Processing 3D medical image data formats includes converting the provided input data from labelled medical data formats into simple stack of images and can be a confusing task for someone who does not know any details of different medical formats. Medical image represents an internal structure of an anatomic region in the form of an array of elements called pixels in 2D or voxels in 3D. It is a result of a sampling/reconstruction process mapping numerical values to positions in the space. Medical image data formats include common information such as pixel depth, metadata, pixel data and photometric interpretation. These formats also store more than one 3D data scans and therefore can be used for 4D image processing. All data used in this paper consist of only one 3D scan per patient [18].

Common medical data formats include:

- NRRD—Nearly raw raster data, general medical image data format, data suffixes differ from versions with attached header (.nrrd) and detached header (.raw/.mhd) where metadata are stored separately from the image data.
- Nifti—Neuroimaging Informatics Technology Initiative, this format is usually used for brain imaging data and uses suffix .nii.
- Dicom—Digital Imaging and Communications in Medicine, general image data format and most commonly used for different medical image data, uses suffix .dcm.
- Analyze—Analyze 7.5 format which uses suffixes .img/.hdr, also a detached header format.

An important part of processing medical images is their representation in the 3D vector space. This can be quite challenging and during this process loss of part of the image information can occur. During our data preparation process we converted NRRD 3D scans into image slices and lost the information about image spacing in the third axis. This information had to be recovered and added in order to generate 3D models of the segmented tissue.

### 2.1.2. Brain Dataset

Dataset consisted of 22 MRI T1 weighted brain scans from different patients and each scan contained 257 sagittal slices of human brain. The pictures had resolution $400 \times 400$ pixels and were provided in PNG format. MRI data were provided by the Department of Radiology from The University Hospital Brno and all original slices were labelled by two independent human experts resulting into two sets of ground truth masks suitable for semantic segmentation labeling all pixels either as brain tissue (white) or non-brain tissue (black). Patients involved were in the age of 35–55 years old both male and female and were in good physical condition. Data have been anonymized and approved for research and scientific purposes.

The first expert labelled the data very precisely and accurately and these labels are used as a reference data for training and evaluation. The second expert labelled the data as it is done regularly on everyday basis in medical praxis. These data are used for comparing the algorithm accuracy to accuracy of a human expert segmentation. An example of reference ground truth mask and mask labelled by our system can be seen in Figure 2.

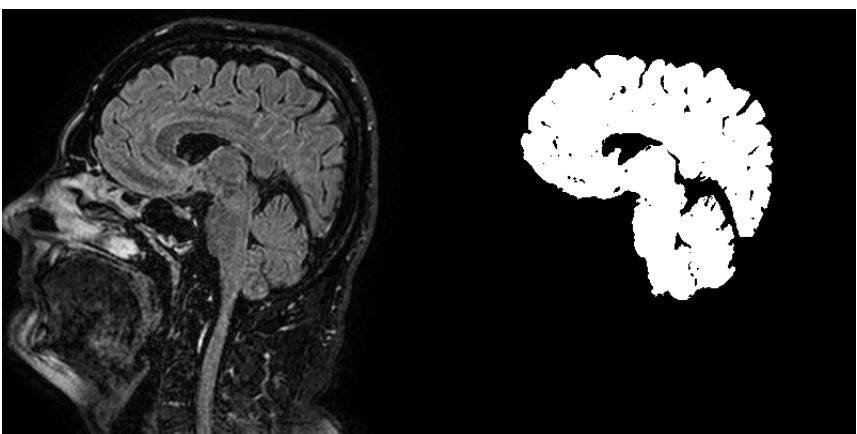

**Figure 2.** Example of brain dataset sagittal image slice (**left**) and according ground truth mask (**right**).

### 2.1.3. Spine Dataset

Spine dataset consists of 10 CT scans of different patients in the age 16–35 years old. The pictures had resolution $512 \times 512$ pixels and were provided in NRRD format. Number of slices in each scan was in range from 520 to 600 slices in the third dimension. Scans cover lumbar and thoracic spine region and were acquired without intravenous contrast. Slice thickness is 1 mm per slice and the in-plane resolution is between 0.31 and 0.45 mm. The data have been acquired at the Department of Radiological Sciences, University of California, Irvine, School of Medicine and scanners used include Philips or Siemens multidetector. Data were published as a part of 2014 CSI workshop challenge of the web http://spineweb.digitalimaginggroup.ca. Dataset can be used for development, training and evaluation of spine segmentation algorithms. Image data are provided in NRRD format. An example image slice and according ground truth mask can be seen in Figure 3.

Ground truth segmentation masks have been semi-automatically segmented and verified for complete thoracic and lumbar vertebrae for each scan. Segmentation masks are also stored in NRRD filed and originally each vertebra had assigned different label. The first vertebra was labelled as 100, the second as 200 and so on. For the semantic binary volumetric segmentation task the masks have been thresholded to greyscale 8bit PNG files and had format where vertebrae tissue was assigned with a value of 255 and non-vertebrae tissue with the value 0 [19].

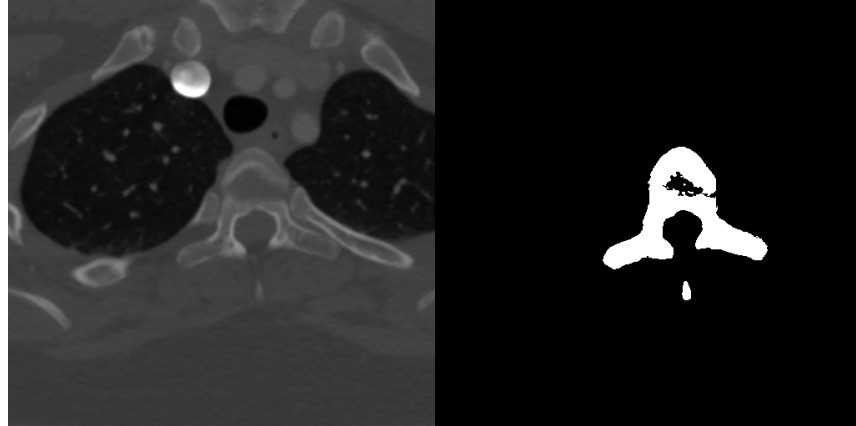

**Figure 3.** Example transversal image slice from spine dataset (**left**) with according ground truth mask (**right**).

### 2.1.4. Denoising Data Preparation

After initial segmentation experiments with the spine dataset we found it is a much more challenging task than the brain volumetric segmentation and training the network only on the dataset was not sufficient for highly precise segmentation. Instead of data augmentation, which for 3D algorithms greatly expands computation time, we used denoising autoencoder pre-training. Principle with this method lies in the fact that we add noise to the training data and let the neural network to learn the representation between noised and denoised data and therefore extracting features of the input image and proper learning of deep layers. An example of original training image, image with added noise and denoised image by our network can be seen in Figure 4.

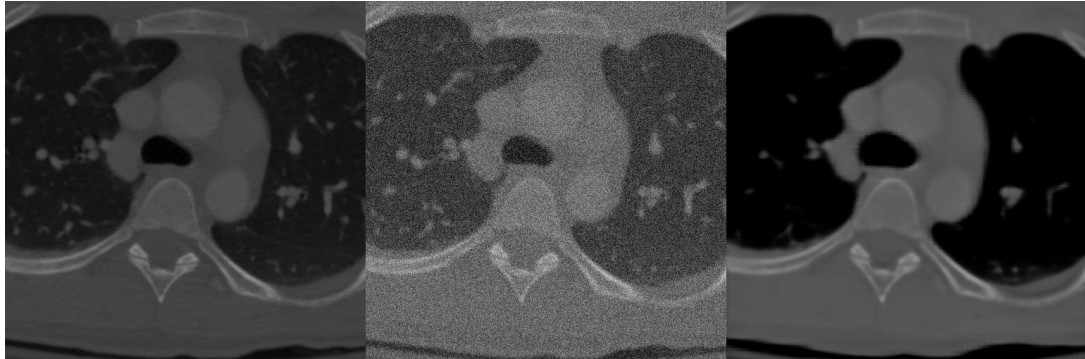

**Figure 4.** Example of data before adding linear noise (**left**), with added linear noise (**middle**) and after denoising by neural network (**right**).

Training dataset consisted of normal training data with added linear 30 percent noise with normal distribution. The network was trained to denoise data into its original form. As seen in Figure 4 the network learned the representation properly, denoised image only lacks higher level of details [20].

### 2.1.5. Training and Testing Data Preparation

Brain dataset consisted of 3D images of brain in resolution $400 \times 400 \times 257$ voxels and spine dataset consisted of 3D thoracic images in resolution $512 \times 512 \times 552$ (some scans have up to 600 images in third axis) voxels. Images of this resolution would not fit into GPU RAM memory (Nvidia GTX 1080ti with 11 GB of RAM) and therefore we had to train our network on smaller batches containing 16 images of brain and for spine only 8 images each. To fully utilize information of 3D data we prepared the input data as overlapping batches. The overlapping technique is used to ensure that the algorithm can utilize as much information over the third axis as possible. If we divided the dataset

into batches of 16 images without overlapping, the training and predictions on the first and the last slice of every batch would not be sufficiently accurate. The algorithm would not have the information of the surrounding slices. Using the batch overlapping technique is ensured that during training every voxel can be analysed using all its surrounding information. The same principle applies for the output prediction of the algorithm. An example can be seen in Figure 5.

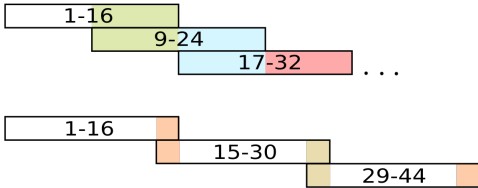

**Figure 5.** Example of training (**upper**) and testing data batch overlapping (**lower**) for brain dataset. Numbers show which slices of the scan each batch contains.

For brain dataset we used 21 scans as a training data, ten percent of that data served for validation and was automatically separated from training dataset by framework Keras [21]. Last remaining 22nd brain scan was used for testing. Due to the fact that approximately 15 slices on both ends of every subject consisted of only non-brain tissue and therefore contained very little information value, we discarded the last slice of every subject and used only 256 slices from each scan for training so the final number is divisible by 16. After the benchmarking process of evaluation of accuracy of used algorithms we trained the fine-tuned Dense-U-Net network three times to ensure generalization of our results using 3 fold cross validation. Dataset has been every round divided into portions of 21 scans for training and 1 for testing.

We wanted to use the same overlapping data technique for outcome predictions from the system. To shorten the time the systems needs to predict the results, we overlapped only 2 pixels on each end of every batch. Prediction then consists of central 14 slices of each batch. This results in 252 testing images used for prediction. An example can be seen in Figure 5.

The spine dataset consisted of ten 3D scanned thoracic CT images. We used nine for training with ten percent of the training dataset used for validation. We used the last remaining scan for testing. After the benchmarking process of evaluation of accuracy of used algorithms we trained the fine-tuned Dense-U-Net network three times to ensure generalization of our results using 3 fold cross validation. Dataset has been every round divided into portions of 9 scans for training and 1 for testing. As the spine segmentation proved to be a more dificult problem than the brain segmentation, we used the same overlapping technique for training and testing datasets. Therefore the neighbouring batches always overlapped four slice each and for prediction on testing dataset we could use inner four slices in each batch and utilize as much information over the third axis as possible.

*2.2. Methodology*

2.2.1. Neural Network Architectures

In this paper we propose 3D Dense-U-Net network which is based on original U-Net implementation [3] and on 3D U-Net version [8] but with added interconnections between layers processing the same feature size, its model is in Figure 6.

We wanted to test the idea of residual and dense interconnections to the segmentation U-net type networks. This principle is used commonly in classification deep neural networks. Using this principle we created and tested the Residual-U-Net and Dense-U-Net networks for 3D data processing which are based on original U-Net and 3D U-Net architectures. Results proved that using residual and dense interconnections can help achieve much better results, but is also computationally expensive both in the terms of time and required GPU memory. The resulting architecture is optimised for high resolution

image processing and can be used on GPU devices with 11 GB of RAM or more. Other work using interconnections in the U-Net [9] use different architecture and use low resolution image processing only. Difference of this work is in the fact that it is capable of processing medical image data in original resolution and achieve higher accuracy than the standard U-Net or 3D U-Net. The interconnections help the network to achieve faster learning curve and obtain higher level of details.

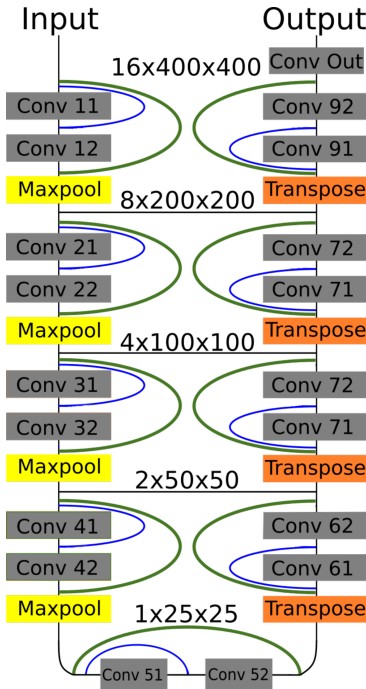

**Figure 6.** Dense-U-Net network model. Residual interconnections are in green color, dense interconnections in blue. Feature sizes are valid for batches of images from brain dataset.

It is an autoencoder type architecture [3] with 4 down-sampling and 4 up-sampling blocks which are connected by a bridge block (in the most lower part of the network). Feature size is halved after passing through every down-sampling block by a maxpooling layer at its bottom. A number of neurons in each layer can be found in Table 1. At the beginning of every up-sampling block the feature size is doubled using transposed convolution layer as described by [22] with stride of size 2 for each dimension. The feature downsampling and upsampling change for four times can only be done fully for the brain dataset, because it has batch containing 16 images. For spine data we have to change the pooling and strides in the most lower parts of the network to the core size of [1,2,2] as can be seen in Table 2. Passing information through interconnections is possible by using zero padding. After every convolution operation we fill the newly computed vector with zeros to its original length and therefore both convolution layers use the same size of the input vector.

**Table 1.** Number of neurons in each layers.

| Input | | Output | |
|---|---|---|---|
| **Layers** | **Neurons** | **Layers** | **Neurons** |
| 11,12 | 32 | 91,92 | 32 |
| 21,22 | 64 | 81,82 | 64 |
| 31,32 | 128 | 71,72 | 128 |
| 41,42 | 256 | 61,62 | 256 |
| 51,52 | 512 | 51,52 | 512 |
| Bridge block—Convolutions 51,52—512 | | | |

Our results in the next section prove, that the 3D Dense-U-Net network exceeds the results of other used architectures. In our experiment we compared the results of original U–Net and Residual-U-Net with added only residual interconnection. On brain dataset we used all three architectures in 3D and their corresponding 2D version to examine whether the 3D models will have better performance. Basic U-Net in 3D and in 2D is an implementation of [3] and Residual-U-Net adds residual interconnections to it (green in Figure 6). Especially Residual-U-Net network had very good results taking into account its number of parameters and can be recommended for application with less computation resources.

**Table 2.** Parameters of network layers—their convolution kernel size, strides and activation function.

| Downsampling Block | | | |
|---|---|---|---|
| **Type** | **Kernel (Pool) Size** | **Strides** | **Activation** |
| Convolution 3D | 3,3,3 | 0,0,0 | Relu |
| Convolution 3D | 3,3,3 | 0,0,0 | Relu |
| Maxpooling 3D | 2(1),2,2 | 2,2,2 | - |
| **Upsampling Block Block** | | | |
| **Type** | **Kernel (Pool) Size** | **Strides** | **Activation** |
| Transpose Conv.3D | 2,2,2 | 2(1),2,2 | - |
| Convolution 3D | 3,3,3 | 0,0,0 | Relu |
| Convolution 3D | 3,3,3 | 0,0,0 | Relu |
| **Bridge Block** | | | |
| **Type** | **Kernel (Pool) Size** | **Strides** | **Activation** |
| Convolution 3D | 3,3,3 | 0,0,0 | Relu |
| Convolution 3D | 3,3,3 | 0,0,0 | Relu |
| **Output Convolution** | | | |
| **Type** | **Kernel (Pool) Size** | **Strides** | **Activation** |
| Convolution 3D | 1,1,1 | 0,0,0 | Sigmoid |

### 2.2.2. Supervised Denoise Pretraining

Rather than to use random values for neural network weights initialization, we pretrained the network first as a denoising autoencoder. The motivation behind this is to help to propagate gradient during the training, help to not get stuck in local minima and increase overall network stability. First some additional noise was added to the input image and then the network was trained to reconstruct the original image. Preparation of the data for this process is described in more detail in Section 2.1.4. In order to make the experiment better comparable over all the examined architectures, 50 epochs limit was used for all the trainings. It resulted in 84.47 percent pixel accuracy in reconstructing the original image (see Figure 4—the network was able to extract the input image features quite successfully). Resulting weights were then used for initialization the fine-tuning phase where 3D Dense-U-Net network was trained using spine dataset. Not only the network started first epoch with validation accuracy over 80 percent (randomly initialized weights have under 30 percent accuracy during first epoch) but also helped the deeper layers to be trained correctly and prevent so called gradient vanishing, which would have significant impact on neural network training capabilities.

### 2.2.3. Training Process

For both datasets all the considered architectures were evaluated based on their computation time and accuracy (see Results in Tables 3–7). They were trained for 50 epochs with the same setting of hyper-parameters. Details regarding hyper-parameter settings can be found in the next Section 2.2.4.

The training of the networks was done in two phases—(1) benchmark phase and (2) fine-tuning phase. First we wanted to compare all the used networks for benchmark purposes to examine whether the Dense-U-Net can achieve the highest accuracy among all the tested architectures. Due to the computation time needed for training we decided to evaluate the tested architectures after 50 epochs of training. When we verified that the Dense-U-Net network performs the best, the limit of 50 epochs was removed and during the fine tuning phase we tried to achieve the best possible accuracy.

Thus the fine-tuned Dense-U-Net network was then trained for 99 epochs using brain dataset. The hyper parameters used were exactly the same as used for the benchmarking phase. This was repeated also for spine dataset using 3D Dense-U-Net network with weights initialized from denoise pre-training. This network was trained as well for 99 epochs. The learning rate was set higher to $10^{-4}$ and the decay to $1.99 \times 10^{-6}$. The reason of that was we wanted to give the network higher learning rate at the beginning, which helped to escape from pre-trained auto-encoder local minimum and to continue on with searching for global minimum.

### 2.2.4. Implementation Details

Complete source code used in this paper is available at [23]. All neural networks architectures were trained using Keras framework [21] using Tensorflow backend. Training was done on Nvidia GTX 1080ti graphics card with 11 GB of memory using CUDA 8.0. As an inspiration for the first U-Net model we used [24] open-source project. Our basic U-Net architecture uses 3D data processing layers. All networks use a binary cross-entropy as a loss function and Adam optimizer with parameters learning_rate equal to $10^{-5}$, *beta*1 to 0.9, *beta*2 to 0.999, *epsilon* to $10^{-8}$ and *decay* to $1.99 \times 10^{-7}$. Using decay parameter we lower the learning rate parameter each epoch by constant value, which helps to fine-tune the network.

Residual-U-Net and Dense-U-Net architectures were designed by adding interconnections to the basic 3D U-Net architecture. Residual-U-Net uses only interconnection over whole down or up-sampling blocks as can be seen in Figure 6 and residual interconnection is denoted as green links. Dense-U-Net uses both residual and dense interconnections (blue in Figure 6) which pass unprocessed information to the middle layer of down and up-sampling blocks.

Network uses as input data values which are in range between 0 and 1. For this reason the input pixel or voxel values, which were encoded as 8-bit monochromatic images in png format, i.e., in range [0,255], were normalized to the range 0.0 to 1.0.

As we used sigmoid activation on output network layer, the output of the network is not labeled by just discrete values but with continuous values in range from 0 to 1. For brain dataset we used as only post-processing on predicted data thresholding. All pixels with value lesser than 0.5 were labelled as 0 and with value greater than 1. However the results on spine dataset required thresholding level to be set to 0.9 and above labelled positive, under the level negative. This has to be done to ensure least amount of artifacts in the output segmentation. Even after thresholding usually one middle sized artifact in the upper thoracic region stayed in the segmented image. To ensure clean segmentation suitable for 3D modeling we had to threshold the model to remove all stand alone objects smaller than 7500 voxels. This ensured the quality output without any artifacts in the segmented image.

### 3. Results

Five different metrics for evaluation were used so the results can be easily compared to related works. All metrics measured are compared to masks labelled by human expert which are used as ground truth. The first used metrics is pixel accuracy, its equation is (1) where $N_{\text{TP}}$ stands for true positive pixels or voxels, $N_{\text{TN}}$ true negative, $N_{\text{FP}}$ false positive and $N_{\text{FN}}$ false negative.

$$A_P(X, Y) = \frac{N_{\text{TP}} + N_{\text{TN}}}{N_{\text{TP}} + N_{\text{TN}} + N_{\text{FP}} + N_{\text{FN}}} \tag{1}$$

$$D(X,Y) = \frac{2 * |X \cap Y|}{|X| + |Y|} \tag{2}$$

$$I_{oU}(X,Y) = \frac{|X \cap Y|}{|X \cup Y|} = \frac{|X \cap Y|}{|X| + |Y| - |X \cap Y|} \tag{3}$$

Dice coefficient [25] is expressed by Equation (2) and intersection over union [26], also known as Jaccard index, is expressed by Equation (3). *X* and *Y* stand for a set of positive pixels/voxels on first and second compared mask. We used Visceral segmentation tool [26] for computing the results in all metrics.

The training of the segmentation neural networks used in this paper is relatively computationally demanding. For this reason, we compare the resulting architectures also from the point of view of computational time needed for training and computational time needed for prediction. Results can be seen in Table 3. As it is obvious from the table, the computational time increases with the complexity of the model and image data resolution, on the other hand it is in all the cases below one minute during application of the model to data.

**Table 3.** Segmentation alogorithms comparison based on their computation time when ran on gtx 1080ti GPU.

| Segmentation Algorithm | Prediction Time | Training Time [50 epochs] |
|---|---|---|
| U-Net—brain dataset | 8 s | 6 h |
| U-Net—spine dataset. | 23 s | 9 h |
| Res-U-Net—brain dataset | 12 s | 10 h |
| Res-U-Net—spine dataset | 32 s | 16 h |
| Dense-U-Net—brain dataset | 21 s | 12 h |
| Dense-U-Net—spine dataset | 43 s | 21 h |
| FSL—brain dataset | 3 s | - |

### 3.1. Brain Dataset Results

First all the architectures were evaluated on brain dataset. The considered architectures are Dense-U-Net, Residual-U-Net and basic U-Net network, all were tested in 2D and 3D mode, i.e., six neural network architectures. This phase verified, that 3D Dense-U-Net architecture performs better than the remaining architectures (see Table 4).

**Table 4.** Comparison of tested U-Net versions on brain dataset in benchmark training phase versus Human expert and FMRIB's Automated Segmentation Tool. Used metrics—pixel accuracy, Dice coefficient, intersection over union, average Hausdorff distance [voxel] and area under ROC curve.

| 3D Networks | | | | | |
|---|---|---|---|---|---|
| Metric | Dense-U-Net | Res-U-Net | U-Net | Human | FSL |
| P.A. | 0.99703 | 0.99662 | 0.99619 | 0.99489 | 0.94289 |
| Dice c. | 0.98843 | 0.98686 | 0.98514 | 0.98033 | 0.79698 |
| I.o.U. | 0.97713 | 0.97407 | 0.97072 | 0.96141 | 0.66248 |
| A.H.D. [voxel] | 0.01334 | 0.01911 | 0.02427 | 0.02479 | 4.58848 |
| A.u.R.C. | 0.99439 | 0.99353 | 0.99205 | 0.98325 | 0.96696 |
| 2D Networks | | | | | |
| Metric | Dense-U-Net | Res-U-Net | U-Net | Human | FSL |
| P.A. | 0.99576 | 0.99639 | 0.99636 | 0.99489 | 0.94289 |
| Dice c. | 0.98344 | 0.98357 | 0.98574 | 0.98033 | 0.79698 |
| I.o.U. | 0.96743 | 0.96768 | 0.97189 | 0.96141 | 0.66248 |
| A.H.D. [voxel] | 0.09477 | 0.09130 | 0.05632 | 0.02479 | 4.58848 |
| A.u.R.C. | 0.99647 | 0.99639 | 0.99663 | 0.98325 | 0.96696 |

As stated in Section 2.1.5, the performance was evaluated on the last brain scan which was not used for training and therefore our results show reliability of our method on unseen data. We compared the predicted data to mask labelled by a human expert as in standard medical praxis, which are in Table 4 in column labelled "Human" and also to output of the FSL segmentation tool.

Dense-U-Net network proved to give the best performance in all considered metrics in 3D processing mode. There should be stated, that even segmentation using basic U-Net network gave better accuracy than a human expert. When compared to other 2D networks, the Dense-U-Net had better results in all metrics except for average Hausdorff distance. This is most probably caused by the fact that 50 epochs of training was not enough for a network with so many parameters and therefore the network generated more artifacts in the output segmentation than the simpler versions. As can be seen in Figure 7 the methods had problems segmenting the area around nasal cartilage. FSL method results in this case were unusable for medical praxis. Dense-U-Net network had the best results also thanks to the fact, that it was able to successfully segment this area on the MRI scan.

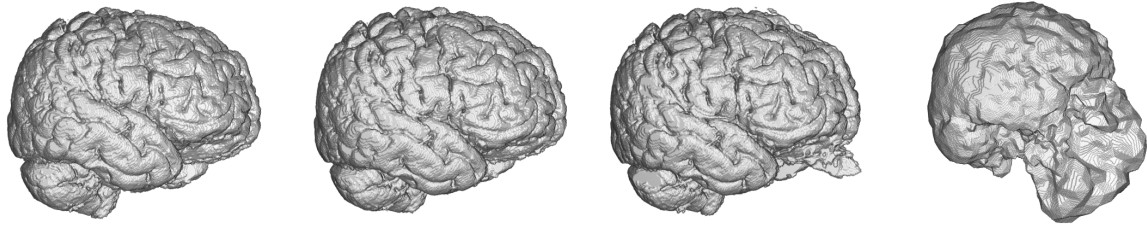

**Figure 7.** Visualisation of predictions from 3D benchmark brain models and FSL segmentation. From left to right—Dense-U-Net, Residual-U-Net, U-Net, FSL.

After the evaluation in the benchmark phase we trained the 3D Dense-U-Net again, now with 99 epochs, to get the final fine-tuned model. Obtained results are in Table 5.

**Table 5.** Fine-tuned model trained using 3-folds cross-validation, results includes standard deviation. Used neural network is Dense-U-Net network. Achieved results are compared to another human expert results. For evaluation, several metrics were used: pixel accuracy, Dice coefficient, intersection over union, average Hausdorff distance [voxel] and area under ROC curve.

| Metric | Dense-U-Net | Human |
|---|---|---|
| P.A. | $0.99721 \pm 0.00026$ | $0.99510 \pm 0.00032$ |
| Dice c. | $0.98870 \pm 0.00134$ | $0.98087 \pm 0.00167$ |
| I.o.U. | $0.97873 \pm 0.00165$ | $0.96032 \pm 0.00189$ |
| A.H.D. [voxel] | $0.01302 \pm 0.00183$ | $0.02519 \pm 0.00342$ |
| A.u.R.C. | $0.99463 \pm 0.00022$ | $0.98413 \pm 0.00093$ |

The Dense-U-Net network was trained in the fine-tuning phase (99 epochs) and validated using 3-folds cross-validation. In all five metrics the results are more accurate than a human expert. Output of the segmentation is visualised in Figure 8. It is clearly visible, that the proposed segmentation output overcame the human results and also results of older methods not based on deep neural networks such as the FSL. The network was able to learn the input image features as well as to generalize the brain segmentation problem. All the evaluations were made using data independent to training and the validation data.

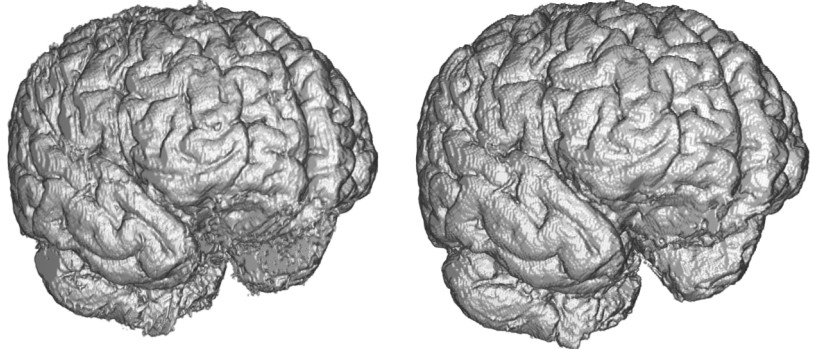

**Figure 8.** Comparison of ground truth brain model (**left**) and brain model segmented by Dense-U-Net (**right**) after the final phase of training.

## 3.2. Spine Dataset Results

As the benchmark comparison of tested architectures in 2D vs. 3D versions is done on brain dataset, we trained and evaluated only 3D versions of the networks. A comparison of results achieved during benchmark phase using all three architectures can be seen in Table 6 and visualisation of the segmented spine models can be seen in Figure 9. This dataset does not contain second set of labels done by a human expert and therefore the results cannot be compared with human precision.

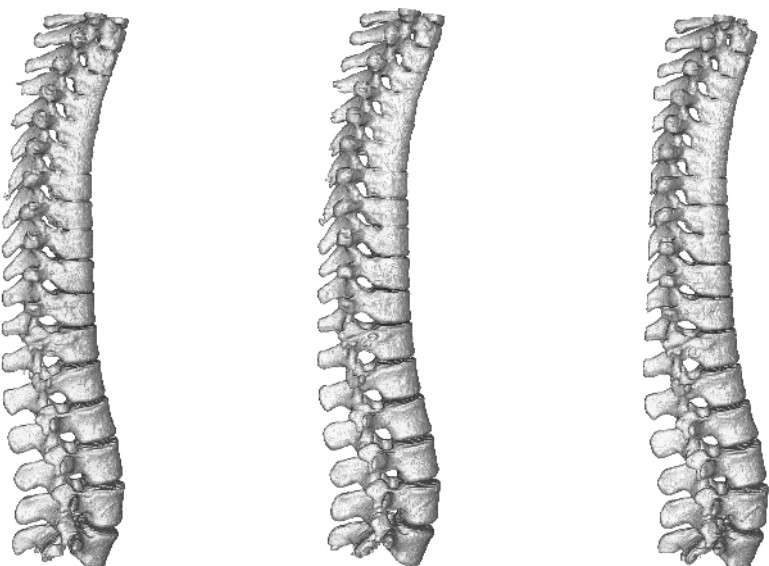

**Figure 9.** Visualisation of segmented spine models from the benchmark training phase. From left to right—Dense-U-Net, Residual-U-Net, U-Net. Please notice that the abnormal vertebrae adhesions exist also in original ground truth masks as can be seen in Figure 10.

**Table 6.** Comparison of tested 3D U-Net versions on spine dataset in benchmark training phase. The training was limited by 50 epochs for each. Used metrics for evaluation are: pixel accuracy, Dice coefficient, intersection over union, average Hausdorff distance [voxel] and area under ROC curve.

| Metric | Dense-U-Net | Res-U-Net | U-Net |
|---|---|---|---|
| P.A. | 0.99732 | 0.99727 | 0.99721 |
| Dice c. | 0.96784 | 0.96733 | 0.96635 |
| I.o.U. | 0.93770 | 0.93672 | 0.93490 |
| A.H.D. [voxel] | 0.21982 | 0.08754 | 0.09226 |
| A.u.R.C. | 0.98589 | 0.98539 | 0.98262 |

Dense-U-Net network has achieved the highest results in all metrics except for average Hausdorff distance. Reason for this is that the model does not perform well on borders of the image. In the dataset there were labelled only thoracic and lumbar vertebrae, but the CT scans contained also the first cervical vertebra and the network did include it in its segmentation results. You can see a part of the first cervical vertebra on top of the segmented spine in Figure 10.

After accuracy of Dense-U-Net was verified to outperform the other architectures in the benchmark phase, we trained the network in the fine-tuning phase to achieve the best results possible, now using 99 epochs with weights initialized using pre-trained model. 3-folds cross-validation was used for evaluation. Results of the Dense-U-Net network in fine-tuning phase is depicted in Table 7. The visualisation of fine-tuned Dense-U-Net network result can be seen in Figure 10. Please notice that the abnormal vertebrae adhesions exist also in ground truth masks on the model in Figure 10 and therefore it is not a failure of the segmentation algorithm.

Results on testing data which the network had not seen during the training process show, that the algorithm is capable of precise segmentation of human spine on CT images. The network has achieved on the spine dataset even better results in metrics pixel accuracy and in dice coefficient in comparison with volumetric segmentation of MRI brain images. This is a very good result because of much higher difficulty of the volumetric CT spine segmentation problem.

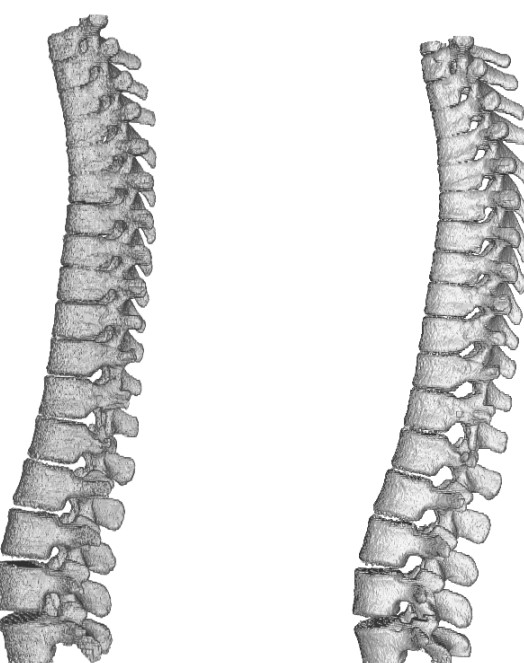

**Figure 10.** Comparison of ground truth model (**left**) and model segmented by Dense-U-Net network (**right**) after the fine-tuning phase of training. Notice difference on top of the figure—the first cervical vertebra and abnormal vertebrae adhesions that exist also in the original ground truth mask.

**Table 7.** Segmentation results using Dense-U-Net and spine dataset in the fine-tuning phase. Results includes standard deviation of 3-folds cross-validation. Used metrics are: pixel accuracy, Dice coefficient, intersection over union, average Hausdorff distance [voxel] and area under ROC curve.

| Metric | Dense-U-Net |
|--------|-------------|
| P.A. | $0.99805 \pm 0.00014$ |
| Dice c. | $0.97082 \pm 0.00292$ |
| I.o.U. | $0.94332 \pm 0.00553$ |
| A.H.D. | $0.16711 \pm 0.11010$ |
| A.u.R.C. | $0.98627 \pm 0.00220$ |

## 4. Conclusions

In this paper we have proposed 3D Dense-U-Net: a new upgraded U-Net architecture with densely connected layers optimized for high resolution 3D medical image data analysis. We evaluated the performance of the network on two independent datasets. The first dataset is the MRI T1 brain dataset and this network achieved pixel accuracy on testing data $99.72 \pm 0.02$ percent which exceeded human expert performance done as in standard medical praxis ($99.51 \pm 0.03$ percent). On the second spine dataset the network achieved $99.80 \pm 0.01$ percent accuracy, which surpassed its results on brain dataset. The network can segment high resolution 3D data in less than one minute using standard PC equipped with Nvidia GTX 1080ti. Using data preparation technique described in Section 2.1.5, we were able to analyse data with a deep neural network for 3D data segmentation using GPU with 11 GB RAM in its original resolution. Many other related segmentation algorithms are designed for data in smaller resolution (because of time or memory demands) and therefore the results are often not usable in practice.

Our approach can easily be applied to any segmentation method already using U-Net architecture. Resulting source-code was released as an open-source and its link is provided in Section 2.2.2. In Appendix A. we provide a manual how to use the source code for other data.

In future we plan to further upgrade data preparation technique so we will be able to train more densely interconnected architectures, plan to evaluate our method on other publicly available datasets and to design a more universal data preparation technique. Also we plan to learn the network using denoising training in a semi-supervised manner to detect novelty in image dataset.

**Author Contributions:** Data curation, K.Ř.; Formal analysis, M.K.D.; Methodology, M.K., R.B., V.U. and M.K.D.; Software, M.K. and V.U.; Supervision, R.B.; Visualization, K.Ř.; Writing—original draft, M.K.

**Funding:** This research was funded the Ministry of the Interior of the Czech Republic by the grant VI20172019086, Ministry of Health of the Czech Republic by the grant NV18-08-00459 and the National Sustainability Program under grant LO1401.

**Acknowledgments:** Research described in this paper was financed by the Ministry of the Interior of the Czech Republic by the grant VI20172019086 and the National Sustainability Program under grant LO1401. For the research, infrastructure of the SIX Center was used. This work was also supported by Ministry of Health of the Czech Republic, grant nr. NV18-08-00459.

**Conflicts of Interest:** The authors declare no conflict of interest.

## Abbreviations

The following abbreviations are used in this manuscript:

| | |
|---|---|
| GPU | graphic processing unit |
| MRI | magnetic resonance imaging |
| CT | computed tomography |
| NRRD | nearly raw raster data medical image format |
| FSL | FMRIB's Automated Segmentation Tool |

## Appendix A

Source code consists of two main files—"data.py" for data loading and preprocessing and "dense-unet.py", which trains the network and segment data and evaluate performance. Depending on the size of the input data you should accordingly modify dimension for resolution and batch depth in data.py. This information has to be changed in dense-unet.py as well. Code is aimed to load slices of .png images located in folders /train, /masks, /test and to create numpy files with encoded images. These files can be used to train the network and make prediction using functions train() and predict() in the dense-unet.py. We recommend using anaconda virtual python environment for execution, the source file for its creation is also included in the github repository.

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
