# Peer review of "Optimized High Resolution 3D Dense-U-Net Network for Brain and Spine Segmentation"

_applsci, doi:10.3390/app9030404_

Round 1

Reviewer 1 Report

The paper presents an expansion of the 3D U-Net architecture for the task of brain and spine segmentation. The approach uses medical images in their original resolution and adds densely connected layers to the U-Net architecture. The author compares their proposed method to the U-Net and Residual-U-Net and produces improved results.

Comments

1.     Residual-U-Net adds residual interconnection to the U-Net architecture and Dense-U-Net adds both residual and dense interconnection to the U-Net architecture. In Figure.6, the only difference between the residual and dense interconnection is the location of the interconnection, which is also very similar, so the novelty of the manuscript is not very convincing.

2.     In experiments, only one scan was used for testing, which was not sufficient compared to the training set. A fair test would be to select more scans as testing set or use 5 fold cross validation. More methods should be compared, and the experiment results do not make enough improvement to provide the reader with a motivation to use the approach.

3.     In Fig.7 and Fig.8, there were only ground truth and the model segmented by Dense-U-Net, predictions from the other two approaches should also be compared, which could help to understand the difference among these networks.

4.     In section 3.2 “Spine dataset results”, in order to improve the performance of Dense-U-Net, the author trained it for 99 epochs with weights initialized from denoise pre-training process, and the related result was shown in Table.6. It was not a fair comparison, as the other two models did not adopt the same training way or use the weights initialized from denoise pre-training process.

5.     There was an error in the number of figures, in the line 319 and 320, I suppose that it should be Fig.6 rather than Fig.4.

6.     Some sentences are too long to read and understand, such as the first sentence in Abstract.

Author Response

Response to Reviewer 1 Comments

Point 1: Residual-U-Net adds residual interconnection to the U-Net architecture and Dense-U-Net adds both residual and dense interconnection to the U-Net architecture. In Figure.6, the only difference between the residual and dense interconnection is the location of the interconnection, which is also very similar, so the novelty of the manuscript is not very convincing.

Response 1: We agree. This has not been clearly explained. The purpose of the paper is to improve the U-Net results using residual and dense principle interconnections which has not been done in this form. Both tested Residual and Dense U-net architecture versions are proposed in this paper. Only work using densely connected network of U-Net type is cited but the implementation in the cited work is different and is used for low resolution image processing. We added more information on this point in subsection 2.2.1. 

Point 2: In experiments, only one scan was used for testing, which was not sufficient compared to the training set. A fair test would be to select more scans as testing set or use 5 fold cross validation. More methods should be compared, and the experiment results do not make enough improvement to provide the reader with a motivation to use the approach.

Response 2: We agree. We computed 3 fold cross validation on both final training phases for brain and for spine. The 3 fold cross validation was used taking into account the computiational time. We also added FSL brain method segmentation for comparison, which is a verified non DNN method for brain matter segmentation-

Point 3:  In Fig.7 and Fig.8, there were only ground truth and the model segmented by Dense-U-Net, predictions from the other two approaches should also be compared, which could help to understand the difference among these networks.

Response 3: We agree. We added Fig. 7 and Fig. 9 to visualise results from all three networks and for brain also the results of FSL segmentation.

Point 4: In section 3.2 “Spine dataset results”, in order to improve the performance of Dense-U-Net, the author trained it for 99 epochs with weights initialized from denoise pre-training process, and the related result was shown in Table.6. It was not a fair comparison, as the other two models did not adopt the same training way or use the weights initialized from denoise pre-training process.

Response 4: We agree. This has not been clearly explained. The training and evaluation was done in two phases - the benchmark and final stage. in the benchmark phase we evaluated the networks trained for 50 epochs to prove the Dense-U-Net has the best results. In the final phase we trained the networks to achieve the best results possible. To achieve the best results we decided to use the denoise pretraining after we have taken into account the results in the benchmark phase. We added paragraph to subsection 2.2.3 to explain point for further readers.

Point 5: There was an error in the number of figures, in the line 319 and 320, I suppose that it should be Fig.6 rather than Fig.4.

Response 5: We agree. Error has been corrected.

Point 6: Some sentences are too long to read and understand, such as the first sentence in Abstract.

Response 6: We agree. We rewrote the abstract to be more clearly written and tried to correct this problem where we found it in the rest of the text.

Reviewer 2 Report

This paper proposes an automated image segmentation method of brain region from T1* MR images (22 subjects), and spine region from CT images (10 subjects). It achieved 99.70 % accuracy in brain segmentation, and 99.79 % accuracy in spine segmentation. 

(1) Sec 1.1.1 should be omit because it describes basic knowledge of medical images, and image processing techniques. The footnote of Sec 2 and Sec. 2.1.1 should be omit. 

(2) In Sec. 2.1.2 and 2.1.3, the authors should describes image acquisition parameters, and patient history (Sex, age, disease, etc). And, the authors must describe ethics committee approval. 

(3) As the authors mentioned at line 136-140, Ref. 9 and 10 proposed 3D U-Net. The authors should discuss the limitation of these methods, and the novelty of this paper in comparison with them. Originality of this paper should be modification of 3D U-Net. However, modification described in Sec. 2.2.1 is very minor change. The author should strength this section.

(4) In Sec. 2.1.5, the method employs the batch overlapping, Explain the advantage of batch overlapping. 

(5) In Sec 3, the authors should describe the computation time, and compare with the existing methods because the strong point of this paper might be computation time as the authors mentioned.

(6) In Table 5, the authors compared the results with existing methods. In order to validate the performance of the modified 3D U-Net, denoising pre-processing should be applied before applying the existing methods.

(7) Development of the brain segmentation has long history. The authors should compare with the other methods (e.g., SPM, FSL, etc) that donot use DNN.  Also, because the number of subjects is few in comparison with the other papers, we cannot compare the performance directly. 

Author Response

Response to Reviewer 2 Comments

Point 1: Sec 1.1.1 should be omit because it describes basic knowledge of medical images, and image processing techniques. The footnote of Sec 2 and Sec. 2.1.1 should be omit. 

Response 1: We agree. We omitted both the Sec. 1.1.1 and the footnote.

Point 2: In Sec. 2.1.2 and 2.1.3, the authors should describes image acquisition parameters, and patient history (Sex, age, disease, etc). And, the authors must describe ethics committee approval. 

Response 2: We agree. We added this information to the Sec. 2.1.2 and 2.1.3.

Point 3:  As the authors mentioned at line 136-140, Ref. 9 and 10 proposed 3D U-Net. The authors should discuss the limitation of these methods, and the novelty of this paper in comparison with them. Originality of this paper should be modification of 3D U-Net. However, modification described in Sec. 2.2.1 is very minor change. The author should strength this section.

Response 3: We agree. We added paragraph explaining the difference between these works and the submitted paper to Sec. 2.2.1.

Point 4: In Sec. 2.1.5, the method employs the batch overlapping, Explain the advantage of batch overlapping.

Response 4: We agree. We added a paragraph explaining the advantage of the method into section 2.1.5.

Point 5: In Sec 3, the authors should describe the computation time, and compare with the existing methods because the strong point of this paper might be computation time as the authors mentioned.

Response 5: We agree. We added a table with measured computation time for training and predictions to the section 3.

Point 6: In Table 5, the authors compared the results with existing methods. In order to validate the performance of the modified 3D U-Net, denoising pre-processing should be applied before applying the existing methods.

Response 6: We agree. This has not been clearly explained. The training was done in two phases - benchmark and final phase. The networks were first trained for 50 epochs in the benchmark phase to prove that the Dense-U-Net will achieve the best results. After this has been proven we wanted to train it to obtain the best results possible in the final phase. We added a paragraph to Sec. 2.2.3 explaining the training process.

Point 7: Development of the brain segmentation has long history. The authors should compare with the other methods (e.g., SPM, FSL, etc) that donot use DNN.  Also, because the number of subjects is few in comparison with the other papers, we cannot compare the performance directly. 

Response 7: We agree. We computed the segmentation using the FSL method and added the results to section 3.1.

Round 2

Reviewer 1 Report

All of my concerns have been addressed. Further proofreading and re-organization of the sentences are encouraged for better readability.

Author Response

Point 1: All of my concerns have been addressed. Further proofreading and re-organization of the sentences are encouraged for better readability.

Response 1: We agree - we performed extensive proofreading and text re-organization to to ensure better readability.

Reviewer 2 Report

Authors revised the paper according to reviews’ comment properly. No additional comments.

Author Response

Point 1: Authors revised the paper according to reviews’ comment properly. No additional comments.

Response 1: Thank you - we still performed extensive proofreading and text re-organization to to ensure better readability.